# The Sixth Edition of the WHO Manual for Human Semen Analysis: A Critical Review and SWOT Analysis

**DOI:** 10.3390/life11121368

**Published:** 2021-12-09

**Authors:** Florence Boitrelle, Rupin Shah, Ramadan Saleh, Ralf Henkel, Hussein Kandil, Eric Chung, Paraskevi Vogiatzi, Armand Zini, Mohamed Arafa, Ashok Agarwal

**Affiliations:** 1Department of Reproductive Biology, Fertility Preservation, Andrology and CECOS, Poissy Hospital, 78300 Poissy, France; florenceboitrelle@yahoo.fr; 2Department BREED, UVSQ, INRAE, Paris Saclay University, 78000 Jouy-en-Josas, France; 3Division of Andrology, Department of Urology, Lilavati Hospital and Research Centre, Mumbai 400050, India; rupinurvashishah@gmail.com; 4Department of Dermatology, Venereology and Andrology, Faculty of Medicine, Sohag University, Sohag 82524, Egypt; salehr2010@yahoo.com; 5Ajyal IVF Center, Ajyal Hospital, Sohag 82524, Egypt; 6Department of Metabolism, Digestion and Reproduction, Imperial College London, London SW7 2AZ, UK; ralf.henkel@logixxpharma.com; 7Department of Medical Bioscience, University of the Western Cape, Cape Town 7535, South Africa; 8American Center for Reproductive Medicine, Cleveland Clinic, Cleveland, OH 44195, USA; mohamedmostafaarafa@gmail.com; 9LogixX Pharma, Theale, Reading, Berkshire RG7 4AB, UK; 10Fakih IVF Fertility Center, Abu Dhabi 31452, United Arab Emirates; hkandil@gmail.com; 11AndroUrology Centre, Brisbane, QLD 4230, Australia; ericchg@hotmail.com; 12Department of Urology, Princess Alexandra Hospital, University of Queensland, Brisbane, QLD 4120, Australia; 13Andromed Health & Reproduction, Fertility Diagnostics Laboratory, Maroussi, 15126 Athens, Greece; evivogiatzi@gmail.com; 14Department of Surgery, McGill University, Montreal, QC H3A 1G5, Canada; ziniarmand@yahoo.com; 15Andrology Department, Cairo University, Giza 11562, Egypt; 16Urology Department, Hamad Medical Corporation, Doha P.O. Box 3050, Qatar

**Keywords:** WHO laboratory manual 6th Edition, semen, sperm, sperm DNA fragmentation, oxidative stress, SWOT

## Abstract

Semen analysis is the cornerstone of male fertility evaluation with WHO guidelines providing the basis for procedural standardization and reference values worldwide. The first WHO manual was published in 1980, and five editions have been subsequently released over the last four decades. The 6th Edition was published in July 2021. In this review, we identify the key changes of this 6th Edition. Additionally, we evaluate the utility of this 6th Edition in clinical practice using SWOT (strengths, weaknesses, opportunities, and threats) analysis. This new Edition has made the analysis of basic semen parameters more robust, taking into account the criticisms and grey areas of the previous editions. The tests assessing sperm DNA fragmentation and seminal oxidative stress are well-described. The main novelty is that this latest edition abandons the notion of reference thresholds, suggesting instead to replace them with “decision limits”. While this seems attractive, no decision limits are proposed for either basic semen parameters, or for extended or advanced parameters. This critical review of the 6th Edition of the WHO laboratory manual combined with a SWOT analysis summarizes the changes and novelties present in this new Edition and provides an in-depth analysis that could help its global use in the coming years.

## 1. Introduction

Infertility is defined as the “inability to achieve spontaneous pregnancy within one year of regular unprotected sexual intercourse” [1]. It is estimated to affect between 48 million couples and 186 million individuals globally [2,3]. Infertility affects around one in eight couples of reproductive age, with a male factor being solely responsible in 20% and contributory in an additional 30% of cases [4]. Hence, a male factor could be present as a primary or contributing cause in approximately 50% of couples [5,6]. In recent years, there has been growing concern about the declining sperm concentrations around the world, which could be attributed to various lifestyle factors such as obesity and exposure to environmental chemicals/radiations [7].

Semen analysis (SA) represents the most basic evaluation of male infertility. The evaluation of semen parameters is currently based on the standards defined in the laboratory manual for the examination and processing of human semen created by the World Health Organization (WHO) [8,9,10]. From the 1st Edition of the WHO laboratory manual published in 1980 to the current 6th Edition [11], there have been significant advances with the incorporation of recent developments in semen examination techniques, methods of sperm preparation and cryopreservation, and new technologies to improve quality control and assurance [12]. Recent scientific advances in the understanding of sperm DNA fragmentation (SDF), seminal oxidative stress (OS), and reactive oxygen species (ROS) testing have shed additional light on the prognosis of reproductive outcomes in terms of natural conception and assisted reproductive technology (ART) [13]. Given the growing awareness that chromosomal abnormalities and gene mutations often underlie a diverse spectrum of male infertility, genetic and genomic testing are gaining attention in this new 6th Edition (see chapter “Extended examinations”). In the chapter “Advanced examinations”, some other tests used in research are described, such as sperm acrosome reaction, functional analysis of transmembrane ion flux and transport in sperm, and methods for the evaluation of chromatin condensation. On the other hand, older tests such as cervical mucus examination have been removed from this new manual.

In this review, we recognize the key changes and new recommendations of the 6th Edition and discuss the objectives and methodological aspects of these changes. Additionally, we utilize a SWOT (strengths, weaknesses, opportunities, and threats) analysis to highlight the merits as well as the limitations of this new 6th Edition, in the context of clinical practice and to provide insights into further steps to improve the manual and optimize its role in the management of infertility worldwide.

## 2. Critical Review of the 6th Edition

### 2.1. Methodological Considerations in the New 6th Edition

Since the release of the 1st Edition in 1980, the WHO Laboratory Manual for Examination of Human Semen had, as its main objective, the standardization of laboratory procedures for human semen examination. From 1980 and onwards, the WHO pursued a uniform approach with the publication of laboratory guidelines and thresholds while periodically incorporating technological advancements and demographic evidence to profile the global male population adequately.

The 5th Edition, in 2010, aimed to facilitate the standardization of SA procedures through a detailed step-by-step approach to various basic and optional semen tests [14]. The 5th Edition, also included a comprehensive part on cryopreservation, which plays an important role in fertility preservation and ART. Another important addition was sperm processing for testicular and epididymal sperm, where standardization promoted better handling across the clinical andrology and ART laboratories. Quality assurance protocols were proposed to ensure strict adherence to the proposed methodologies and reporting of SA. Furthermore, laboratory scenarios (so-called worked examples) were presented and explored, along with laboratory troubleshooting to enable greater insight into the practical implementation of the guide.

In general, the 5th Edition added much to the male infertility practice, both from clinical and research perspectives, and the incorporation of data from reference subsets was part of an ongoing attempt to define male fertility in numbers. The reference ranges proposed in the 5th Edition were derived from fertile men whose partners had a time to pregnancy ≤12 months (1959 men from 8 countries representing 4 continents (Europe, Americas and Oceania). These data were combined and analyzed in a multicenter study by Cooper et al. [15] using the 5th centile of one-sided lower reference intervals as an appropriate approach for the classification of semen parameters as fertile or infertile.

However, after its publication, the 5th Edition was criticized for suggesting reference ranges of semen parameters as the mainstay in the evaluation of the male fertility potential [11,16]. The concerns regarding the 2010 reference ranges surrounded its possible inadequacy to represent the general population due to the voluntary nature for the inclusion of most of the cohorts, and the over- and under-representation of some areas of the world and their respective population. Other potential biases included intra and inter-individual biological variations and technical variations owing to intra- and inter-laboratory variation, with some laboratories lacking formal quality assurance and control. Björndahl et al. [17] contemplate the inadequacy of patient selection to represent the general population and suggest using interpretation ranges instead of cut-off limits to assess fertility potential. Additionally, they raise concerns regarding lowering the reference ranges of some parameters and the impact of withholding medical examination in some instances.

The new 6th Edition, released in July 2021, commended the importance of SA as a tool to: (1) assist fertility/infertility diagnosis, (2) assess male reproductive health and function to guide management, (3) guide the choice of ART procedure, (4) monitor response to treatment, and (5) measure the efficacy of male contraception [11]. The 6th Edition aimed to optimize SA procedures by including detailed steps and a methodological sequence for test execution [11]. The newly established manual also described new sperm tests for the assessment of SDF and seminal OS, while abandoning obsolete tests like human cervical mucus. The 6th Edition also aimed to address the drawback of the 5th Edition related to the demographic under-representation of some geographical regions. Hence, the new 6th Edition combined data of the previous 5th Edition and additional new data of fertile men, whose partners had a time to pregnancy ≤12 months, collected between 2010 and 2020 [18]. Thus, the 6th Edition contains results of semen samples of 3589 fertile men (1800 subjects from the 5th Edition and 1789 new subjects). The newly added data originate from two countries in Southern Europe, which were under-represented in the previous 5th Edition, along with two countries from Asia and one country from Africa that lacked representation in the previous release. However, some geographical areas, such as South America and Sub-Saharan Africa, remain under-represented in the current 6th Edition.

An important deviation from the previous 5th version is the abandonment of the reference values. In this 6th Edition, it is clearly specified that the 5th centile values are only one way to interpret the results of SA, and the use of the 5th centile alone is not sufficient to diagnose male infertility. The most notable differences between the 5th [14] and 6th Editions [11] are shown in Table 1.

### 2.2. Basic Examination of Semen

In this 6th Edition, some adjustments have been made regarding basic semen parameters. The evaluation of semen odor has been added, and the manual states that “urine or putrefactive odors may be of clinical interest”. It should be noted that the evaluation of semen odor is subjective [11,19], which makes the standardization of this parameter very complicated. Furthermore, the addition of this parameter is at odds with recommendations for the safety of laboratory personnel. It is possible that some countries will recommend the analysis of this parameter (together with all other sperm parameters), which goes against the rules of personal protection against viruses emerging in the next decade, and whose respiratory transmission could not be excluded.

Concerning sperm motility, the 6th Edition re-adopted the distinction of progressive motility in two categories (grade a and b). Thus, the categorization of sperm motility has reverted to fast progressively motile, slow progressively motile, non-progressively motile and immotile (grade a, b, c, or d, respectively) as mentioned in the 4th Edition [20]. This requires pre-incubation of the clean microscope slides at 37 °C. To justify this choice, the WHO manual cites older papers stating that rapid motility (grade a) has clinical value [21,22,23,24,25,26,27,28,29,30]. Since this assessment was abandoned in the 2010 5th Edition, the distinction of progressive motility in two categories was not evaluated like the other parameters (by comparing the data obtained from men with time to pregnancy ≤12 months with men that remain childless). It is therefore surprising that this distinction of motility is added to the 6th Edition without any recent studies (after 2010) that demonstrate its usefulness in andrology or in routine diagnosis.

For the evaluation of sperm count, semen dilutions have been simplified, but 200 spermatozoa per replicate should be counted. In the past version of the manual, the observation of 0–4 spermatozoa per field at ×400 magnification (or the observation of 0–16 spermatozoa per field at ×200 magnification) could provide enough indication for the assessment of concentration. Indeed, the sperm concentration could be reported as less than 2 × 10^6^/mL. This method has been revised in the 6th Edition. Henceforth, the evaluation of low sperm concentrations (<2 × 10^6^/mL) must be assessed more precisely, by noting that the errors associated with counting a small number of spermatozoa may be very high.

Regarding the indications for sperm vitality assessment, the 6th Edition has addressed an inconsistency surrounding the assessment of sperm vitality that exists in the previous 5th Edition [11,14]. In the previous Edition, if progressive motility was below 40%, vitality testing was recommended, but the 40% threshold itself corresponded to that of total motility. In the 6th Edition, the assessment of sperm vitality is recommended when total sperm motility is below 40%.

Sperm morphology assessment using a systematic approach is described in the new 6th Edition with multiple and better-quality micrographs of spermatozoa from unprocessed semen samples that are considered normal, borderline, or abnormal. These are accompanied by explanations for the classification of each assessment, rendering this a helpful guide. The evaluation of the morphological anomalies of the head, intermediate piece and tail are described. The significance of recording the presence of large cytoplasmic droplets is emphasized.

The thresholds of basic semen parameters used in the 5th Edition, and those described as “useful values” in the 6th Edition, are compared in Table 2. The incorporation of additional areas and continents and, importantly, the addition of more participants and samples in the final analysis provide greater statistical power to the reference ranges reported, even though the 5th centiles are not significantly different from the 2010 WHO 5th Edition [11,14] (Table 2). Notably, throughout the 280 pages of the current Edition, the terms “normozoospermia” “asthenozoospermia”, “necrozoospermia”, “teratozoospermia” are not used at all. These terms have been voluntarily removed as the editors of the manual explain, quite rightly, that the reference thresholds alone are meaningless and that multiple criteria must be applied to establish a diagnosis of male infertility. The latter statement is correct, but in actual practice, it is highly likely that clinicians may encounter some degree of confusion with the absence of reference values. To rely on other reference values, clinicians will need to look for other sources in the literature. This can be time-consuming and difficult. Therefore, a possibility remains that clinicians will continue to use the 5th centile values, which were designed in the 5th Edition to compare fertile and infertile men with the criterion of time to pregnancy ≤12 months.

A feasible solution for such a dilemma is to determine “individualized” reference thresholds or decision limits of SA for selected categories of patients based on a thorough evaluation of their pathology while considering all potential sources of variability in the results. The establishment of clinical decision thresholds, considering certain specific morphological abnormalities as well as racial and ethnic differences, could replace the current 4% threshold. This may help overcome the limitations that surround SA currently and enhance its diagnostic as well as prognostic role in the management of male infertility. Till then, the current threshold issue will remain a problem for laboratories and clinicians.

### 2.3. Extended Examination of Semen

The importance of SDF and genetic evaluation in the context of male infertility has been expanded in the 6th Edition, unlike its previous counterpart, the 5th Edition, which merely described sperm genetics and chromatin evaluation under the research procedures section [11,14]. In addition, the 6th Edition provides a detailed outline on the technical aspects of these tests and some guidance on the interpretation of the test results. However, a discussion on the indications and how to apply the results of these tests in clinical practice is missing.

#### 2.3.1. Sperm DNA Fragmentation

Sperm DNA integrity is a pre-requisite for proper embryo development, implantation, and pregnancy [31] and the editors of the 6th Edition acknowledge that SDF testing “could represent an important addition in the work-up of male infertility, becoming one of the most discussed and promising biomarkers in basic and clinical andrology”. The 6th Edition describes and elaborates on different methods of SDF testing, including the TUNEL assay, sperm chromatin dispersion assay, Comet assay and acridine orange flow cytometry assay. Except for the Comet assay, these tests are deemed useful for clinical testing. However, the 6th Edition provides no thresholds and recommends that each laboratory develop its own reference range based on appropriate controls. Additionally, the clinical utility of SDF testing is not discussed, so the reader is uncertain as to how to use the results in a clinically meaningful way.

The editors of the 6th Edition contributed a section on sperm chromatin integrity testing in the chapter titled “Advanced examinations of semen”. It is unclear why they have chosen to discuss and describe sperm chromatin tests in the chapter on “Advanced examinations of semen” when these tests are closely related to, and certainly no more advanced than tests of SDF.

#### 2.3.2. Genetic and Genomic Tests

In the chapter on “Extended examination of semen”, the editors of the 6th Edition acknowledge the increasing awareness of genetic-related male infertility, specifically, the diverse forms of sperm chromosomal abnormalities and gene mutations [32]. Unselected infertile men, as well as men with SDF, Robertsonian and reciprocal translocation, and those with a history of recurrent pregnancy loss are at increased risk of producing aneuploid sperm [33,34]. The manual addresses the utility of fluorescence in situ hybridization (FISH) as a diagnostic cytogenetic tool in the assessment of chromosomal aberrations and describes the FISH procedure, with emphasis on scoring criteria [35]. FISH is commonly used in the assessment of chromosomal aneuploidy involving chromosomes 13, 18, 21, X, and Y, which usually result in viable but defective births [36]. Additionally, FISH is used in the assessment of the male partner’s chromosomal abnormalities in cases of recurrent pregnancy loss or ART failure [37].

The 6th Edition introduces the reader to sperm genetic tests and how they relate to male reproductive biology and associated fertility disorders, yet it fails to provide guidelines on the indications and utility of these tests. Although the authors list some correlates between abnormal genetic tests and clinical parameters, they do not adequately address the indications for sperm genetic testing. This important omission can lead to inappropriate use of these tests in clinical practice. Moreover, without some guidance as to the utility of these assays, it is unclear how the tests will be used to guide the management of infertile couples. Finally, the 6th Edition acknowledges the scarcity of tools available in the general andrology laboratory and that most tests require an advanced genetic testing facility.

### 2.4. Advanced Examination of Semen

#### 2.4.1. Advanced Tests

Some tests concerning the evaluation of seminal OS, sperm acrosome reaction, functional analysis of transmembrane ion flux, and transport in sperm or methods for the evaluation of chromatin condensation have been described in the 6th Edition.

Concerning the epigenetic mechanisms of chromatin condensation, for example, the aniline blue and chromomycin A3 assays have been described. These assays assess the degree of histone replacement by protamines in the sperm nucleus. However, the actual epigenetic assays, which assess, for example, post-translational histone modifications, the degree of sperm DNA methylation, or small RNA are not described. These assays may have a use in the evaluation of male infertility, however, their adoption by a large number of fertility laboratories will require robust research and clinical trials proving their value in diagnostic andrology [11,38].

It is probably also for this reason that the mitochondrial membrane potential assessment tests are not described in this 6th Edition. If the number of publications on these subjects continues to grow during the next decade, no doubt the next Edition of the WHO manual may mention them.

#### 2.4.2. Seminal Oxidative Stress and Reactive Oxygen Species

The concept of OS was developed by Helmut Sies in 1985 and was subsequently introduced into redox biology and medicine [39]. In the 1990s, Aitken picked up on this concept [40] and introduced it to the field of andrology [41]. Since then, OS been shown to be a major contributing cause of male infertility [42]. Due to the production of excessive amounts of ROS in numerous medical conditions, including varicocele [43], leukocytospermia [44], diabetes mellitus [45], and obesity [46], seminal OS develops. Consequently, OS negatively affects all sperm functions, including sperm DNA integrity [47,48], and thereby the fertilization process and reproductive outcomes [49,50,51,52,53,54,55].

Therefore, it is tempting to use seminal OS as a diagnostic parameter to predict sperm fertilizing potential [56]. In the 5th Edition of the WHO manual, seminal OS has been mentioned under “Research procedures” [14,20]. In the new 6th Edition, seminal OS is described together with ROS in a separate subsection under “Advanced examinations of semen”. Additionally, the determination of oxidation–reduction potential (ORP) using the male infertility oxidative system (MiOXSYS), and total antioxidant capacity are mentioned with a brief description of procedures and issues.

Unfortunately, this 6th Edition of the WHO manual is lacking recently published evidence indicating the predictive power of seminal OS as determined by means of the ORP [48,57,58,59,60]. Similarly, the luminometric determination of seminal OS resulted in a significant prediction of good embryo cleavage and blastocyst quality after intracytoplasmic sperm injection (ICSI) in the low OS group [61]. Earlier, it was shown that the seminal radical buffering capacity is the best predictor of fertilization [62].

Despite more evidence being published recently [55], professional societies have not yet fully endorsed OS as a diagnostic tool. However, as there are a number of different methodologies for the determination of seminal OS with different results described, a consensus on which technique would be best to diagnose male fertilizing potential is lacking. Therefore, the WHO still regards these tests not only as very specialized, but also as mainly research-based or as emerging technologies. In addition, criticism of using OS as a parameter to predict the male fertilizing potential, particularly for the direct determination of either ROS with luminometric methods or ORP using the MiOXSYS System, is also pointing at the pH sensitivity of the measurement, especially at alkaline pH values [48]. In addition, criticism arises due to the fact that ROS are extremely reactive with half-life times in the nano-second range [63] and would therefore be neutralized in the seminal plasma, which is regarded as the human body fluid with the highest amount of antioxidants. Most importantly, however, currently there are insufficient well-designed studies evaluating the impact of seminal OS on the reproductive outcome and providing normal values in order to identify men with abnormal seminal OS levels.

Future good quality studies on the seminal OS are warranted to (1) identify alternative redox parameters [64,65] and (2) improve the diagnostic capabilities of the antioxidant capacity or redox balance by means of ORP. For the latter, recent progress has been made in terms of the identification of relevant patients [60,66,67]. However, at this stage, generally accepted cut-off values for reproductive outcomes are still lacking.

### 2.5. Sperm Preparation and Sperm Cryopreservation

Concerning sperm preparation techniques, the two main novelties of this 6th Edition are the description of the magnetic-Activated cell sorting (MACS) technique and the sperm vitrification technique [11].

In the chapter “Sperm preparation techniques”, this 6th Edition briefly describes the MACS technique as a technique for selecting sperm with potentially undamaged DNA (i.e., with intact DNA). But while the addition of this test represents a novelty, compared to the 5th Edition, this technique is not described as clinically valuable because the WHO states that a Cochrane review has not shown its effectiveness in increasing live birth rates [68]. Furthermore, the MACS technique is not approved for human in vivo use in the United States (and maybe in some other advanced countries as well) due to the lack of safety studies on paramagnetic beads (used in MACS columns to trap sperm with high SDF) in the animal model.

While sperm cryopreservation and sperm preparation techniques are described in a single chapter in the 5th Edition, they are discussed in two separate chapters in the 6th Edition. In the chapter on “Cryopreservation of spermatozoa”, the vitrification technique makes its appearance. Open and closed systems are detailed. However, the manual states that these techniques have not been proven to be superior to conventional freezing techniques and should be considered experimental [11]. Studies on a larger series must be conducted.

### 2.6. SWOT Analysis

The highlights of the 6th Edition are summarized in a SWOT analysis (SWOT stands for “strengths” (S), “weaknesses” (W), “opportunities” (O) and “threats” (T)) (Figure 1). The “strengths” (S) and “weaknesses” (W) are linked to the internal environment of a subject; while the “opportunities” (O) and “threats” (T) are all the external factors that can either enhance or hinder the development of a subject, respectively [69]. The SWOT analysis is a strategic approach to evaluate the 6th Edition of the WHO Manual for Examination of Human Semen. Using this analysis, we summarize the main strengths and weaknesses of the 6th Edition. In addition, we highlight the potential threats that could hinder the global use of this 6th Edition in clinical practice. Furthermore, we provide insights into the available opportunities that can be used to optimize the benefits from this manual as a worldwide reference in human reproduction.

## 3. Conclusions

In this comprehensive review, we conduct an in-depth analysis of the 6th Edition of the WHO procedural manual for human SA, highlighting its strengths, weaknesses, opportunities, and threats. The new 6th Edition presents revised reference values of basic semen parameters based on the combined data of fertile men from the previous 5th Edition, released in 2010, and 5 additional studies published between 2010 and 2020, thereby attempting to address a limitation noted in the 5th Edition due to skewing of the reference values towards normality of specific geolocations. In addition, the 6th Edition incorporates advances in laboratory techniques of semen examination. An example is the introduction of the SDF assay as an extended assessment of semen that can be requested in certain clinical scenarios, although the manual neither provides guidance as to the indication for testing nor addresses the variability of test results with different currently available SDF assays. Additionally, the 6th Edition highlights the recent developments of the techniques of sperm preparation and cryopreservation and recommends that the laboratories adhere to optimum quality control and quality assurance measures. The extensive data on human semen contained in the 6th Edition provide important insights into the global management of male infertility. Expanding our knowledge and understanding of different aspects of human semen will lead to optimizing the care of infertile men and improving the overall reproductive outcome of infertile couples.

## Figures and Tables

**Figure 1 life-11-01368-f001:**
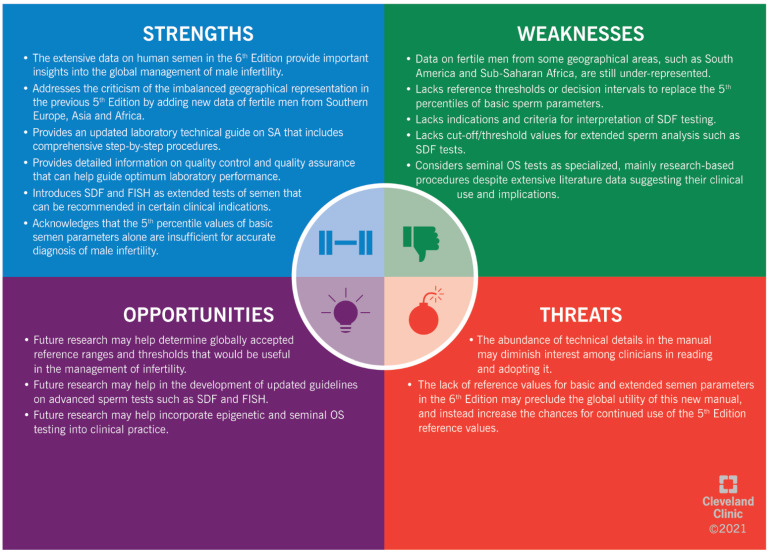
SWOT (“strengths” (S), “weaknesses” (W), “opportunities” (O) and “threats” (T)) analysis of the 6th Edition of the WHO manual for human semen analysis. SA: semen analysis. SDF: sperm DNA fragmentation. FISH: fluorescent in situ hybridization. OS: oxidative stress.

**Table 1 life-11-01368-t001:** Major changes in the objectives and methods from the 5th to the 6th Edition of the WHO Laboratory Manual for the Examination and Processing of Human Semen. SA: semen analysis. ART: assisted reproductive technology. TTP: time to pregnancy. WHO: World Health Organization.

	WHO 5th Edition (2010)	Page Number WHO 5th	WHO 6th Edition (2021)	Page Number WHO 6th
**Objectives**	● Improve the manual by adding detailed description of semen tests and reference ranges	1–3	● Assist fertility/infertility diagnosis	1–2
	● Update SA procedures	1–3	● Assess male reproductive health	1–2
	● Emphasize quality assurance in semen laboratories	179–202	● Guide the choice of ART procedure	1–2
			● Monitor response to treatment	1–2
			● Measure the efficacy of male contraception	1–2
			● Update SA procedures	2–3
			● Eliminate outdated tests	3
**Methods**	● **Reference ranges are provided using 5th centiles based on:**	3	● **Reference ranges and 5th centiles are insufficient to diagnose infertility; 5th centiles are based on:**	4, 211–213
	● Multicenter studies with retrospective data analysis	223–225	● Integration of the 2010 data and reanalysis with data published in the last decade	4, 211–213
	● 1953 men with TTP ≤ 12 months	Cooper et al. [15]	● 3589 men with TTP ≤ 12 months	211–213
	● 8 countries, 4 continents (Oceania, Americas, Europe)	Cooper et al. [15]	● 13 countries, 6 continents (Asia, Americas, Europe, Africa, Oceania)	Campbell et al. [18]

**Table 2 life-11-01368-t002:** WHO 2010 (5th Edition) and WHO 2021 (6th Edition) lower fifth percentile (with 95% confidence interval) of semen parameters from men in couples starting a pregnancy within one year of unprotected sexual intercourse leading to a natural conception.

	WHO 2010	WHO 2021
**Semen volume** (mL)	1.5 (1.4–1.7)	1.4 (1.3–1.5)
**Total sperm number** (10^6^ per ejaculate)	39 (33–46)	39 (35–40)
**Total motility** (%)	40 (38–42)	42 (40–43)
**Progressive motility** (%)	32 (31–34)	30 (29–31)
**Non progressive motility** (%)	1	1 (1–1)
**Immotile sperm** (%)	22	20 (19–20)
**Vitality** (%)	58 (55–63)	54 (50–56)
**Normal forms** (%)	4 (3–4)	4 (3.9–4)

## Data Availability

Not applicable.

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
