# Peer review of "The Sixth Edition of the WHO Manual for Human Semen Analysis: A Critical Review and SWOT Analysis"

_life, 2021, doi:10.3390/life11121368_

Round 1
Reviewer 1 Report
Dear Authors,
The manuscript submitted by Boitrelle F. et al., entitled "The Sixth Edition of the WHO Manual for Human Semen Analysis: A Critical Review and SWOT Analysis," is exciting and brings deep analyses about the changes in the evaluation of human semen analysis in the Sixth Edition of the WHO Manual.
I have only a few suggestions:
- Please include a list of abbreviations in alphabetical order.
- Table 1 - 3 continents? If there were both Americas - 4 continents/6 continents
- l. 165 it should be 106
l. 166 106
Author Response
Manuscript ID: life-1481127 The Sixth Edition of the WHO Manual for Human Semen Analysis: A Critical Review and SWOT Analysis
We would like to thank the reviewers for examining our manuscript in detail and providing insightful comments. Enclosed, please find our point-by-point response to the specific reviewers’ comments.
Reviewer #1:
Dear Authors,
The manuscript submitted by Boitrelle F. et al., entitled "The Sixth Edition of the WHO Manual for Human Semen Analysis: A Critical Review and SWOT Analysis," is exciting and brings deep analyses about the changes in the evaluation of human semen analysis in the Sixth Edition of the WHO Manual.
REPLY: Thank you for your positive comment.
I have only a few suggestions:
1-Please, include a list of abbreviations in alphabetical order.
REPLY: A list of abbreviations has been added to the revised version of the manuscript (inserted before the References section).
2- Table 1 - 3 continents? If there were both Americas - 4 continents/6 continents
REPLY: Thank you. The number of continents has been corrected in the table. Please see Table 1 (under Methods subheading).
3- l. 165 it should be 106; l. 166 106
REPLY: Thank you. These typing errors have been corrected in the revised version. The corresponding lines are lines 185 and 186.
Finally, we would like to thank the reviewers for their very positive and constructive comments. Some typing errors have been corrected (lines 49, 131, 190). We very much hope that our article will be accepted after these revisions.

Reviewer 2 Report
Life
COMMENTS TO THE EDITORS AND THE AUTHORS
Manuscript ID Life-1481127
“The Sixth Edition of the WHO Manual for Human Semen Analysis: A Critical Review and SWOT Analysis “
Dear Editors and the Authors,
Please find enclosed the comments for the above-mentioned manuscript.
A SUMMARY OF THE CONTENT
The authors stated that the review aims to discuss the 6th Edition of the WHO Manual for Human Semen Analysis published in July 2021., to identify key changes of this 6th Edition, as well as to evaluate the utility of the new edition in the clinical practice using SWOT (strengths, weaknesses, opportunities, and threats) analysis. The authors pointed that new Edition has made the analysis of basic semen parameters more robust and that the tests assessing sperm DNA fragmentation and seminal oxidative stress are well-described. The main novelty of the latest edition abandons the notion of reference thresholds, suggesting instead replacing them with “decision limits”. While this seems attractive, no decision limits are proposed for either basic semen parameters or for extended or advanced parameters.
THE OVERALL OPINION OF THE MANUSCRIPT
The manuscript is within the scope of the journal and describes the important topic. The text is comprehensive but very easy to follow. The topic is of interest to general readers. The SWOT analysis is very useful to the readers. The authors have top-rated and well-recognized publications in the field. The authors reference both, pioneered results as well as recent advances in the field. The text and the conclusions support the title.
A few minor suggestions are listed below.
(1) Please consider to comment the small discriminatory range between categories “normozoospermic” and “teratozoospermic” in the 6th Edition of the WHO Manual for Human Semen Analysis published in July 2021.
(2) Please consider to comment that the 6th Edition of the WHO Manual for Human Semen Analysis did not discuss eventual diagnostic potential of mitochondrial membrane potential and other mitochondrial markers (publications availble on PubMed since 2015.).
(3) Please consider to include at the end of the manuscript the paragraph (4) describing the eventual recommendations of the authors for the next edition of the WHO Manual for Human Semen Analysis.
Good luck and all the best :)
Author Response
Manuscript ID: life-1481127 The Sixth Edition of the WHO Manual for Human Semen Analysis: A Critical Review and SWOT Analysis
We would like to thank the reviewers for examining our manuscript in detail, and providing insightful comments. Enclosed, please find our point-by-point response to the specific reviewers’ comments.
Reviewer 2:
Dear Editors and the Authors,
Please find enclosed the comments for the above-mentioned manuscript.
A SUMMARY OF THE CONTENT
The authors stated that the review aims to discuss the 6th Edition of the WHO Manual for Human Semen Analysis published in July 2021., to identify key changes of this 6th Edition, as well as to evaluate the utility of the new edition in the clinical practice using SWOT (strengths, weaknesses, opportunities, and threats) analysis. The authors pointed that new Edition has made the analysis of basic semen parameters more robust and that the tests assessing sperm DNA fragmentation and seminal oxidative stress are well-described. The main novelty of the latest edition abandons the notion of reference thresholds, suggesting instead replacing them with “decision limits”. While this seems attractive, no decision limits are proposed for either basic semen parameters or for extended or advanced parameters.
THE OVERALL OPINION OF THE MANUSCRIPT
The manuscript is within the scope of the journal and describes the important topic. The text is comprehensive but very easy to follow. The topic is of interest to general readers. The SWOT analysis is very useful to the readers. The authors have top-rated and well-recognized publications in the field. The authors reference both, pioneered results as well as recent advances in the field. The text and the conclusions support the title.
REPLY: Thank you for your positive comment.
A few minor suggestions are listed below.
1- Please consider to comment the small discriminatory range between categories “normozoospermic” and “teratozoospermic” in the 6th Edition of the WHO Manual for Human Semen Analysis published in July 2021.
REPLY: Thank you, You are right to point out the difficulties encountered with this threshold by all clinicians in the world. This threshold is the one that stands out in the fifth and sixth editions as being discriminating in the diagnosis of male infertility. We have already stated that "individualized reference thresholds" may need to be established (lines 217-218). We have added (Lines 220-223): “The establishment of clinical decision thresholds, considering certain specific morphological abnormalities as well as racial and ethnic differences, could replace the current 4% threshold. Till then, this threshold issue will remain a problem for laboratories and clinicians.”
2- Please consider to comment that the 6th Edition of the WHO Manual for Human Semen Analysis did not discuss eventual diagnostic potential of mitochondrial membrane potential and other mitochondrial markers (publications availble on PubMed since 2015.).
REPLY: Thank you for your suggestion, we have added a discussion of these and other tests that may be missing in the "advanced examinations" section. This allows us to add our perspective on what might be included in the next Edition, as you suggested in your third comment.
Hence, a sentence has been added to the introduction on this point (lines 65-67)
“In the chapter "advanced examinations", some other tests used in research are described such as sperm acrosome reaction, functional analysis of transmembrane ion flux and transport in sperm, and methods for the evaluation of chromatin condensation.”
And a paragraph has been added at the beginning of the chapter “advanced examinations” (lines 284-297)
“Some tests concerning the evaluation of seminal oxidative stress, sperm acrosome reaction, functional analysis of transmembrane ion flux and transport in sperm or methods for the evaluation of chromatin condensation have been described in this Edition.
Concerning the epigenetic mechanisms of chromatin condensation, for example, the aniline blue and chromomycin A3 assays have been described. These essays assess the degree of histone replacement by protamines in the sperm nucleus. However, the actual epigenetic assays, which assess for example post-translational histone modifications, the degree of sperm DNA methylation or small RNA are not described. These assays may have a use in the evaluation of male infertility, however, their adoption by a large number of fertility laboratories will require robust research and clinical trials proving their value in diagnostic andrology [38].
It is probably also for this reason that the mitochondrial membrane potential assessment tests are not described in this 6th Edition. If the number of publications on these subjects continues to grow during the next decade, no doubt the next Edition of the WHO manual may mention them.”
3- Please consider to include at the end of the manuscript the paragraph (4) describing the eventual recommendations of the authors for the next edition of the WHO Manual for Human Semen Analysis. Good luck and all the best :)
REPLY: Thank you for your suggestion. As this edition has just been published, it seems difficult to determine at this point what could be done better in a future edition, especially if this new edition (the 7th) were to be released in 10 years.
In our review and SWOT analysis, we have detailed what we consider to be opportunities and threats. We have provided in-depth information on what the andrologists/researchers/WHO need to do in the next ten years. Throughout our review, we have tried to give our opinion (based on SWOT analysis) on what could improve the global dissemination of this manual. We preferred to open the debate rather than saying right now that we need another Edition, and what we would like to see changed. It is certain that the global work of the next ten years will improve the seventh Edition. We have given all the advice in our SWOT figure, for the years to come and we prefer to use this figure rather than writing it again in a paragraph.
Finally, we would like to thank the reviewers for their very positive and constructive comments. Some typing errors have been corrected (lines 49, 131, 190). We very much hope that our article will be accepted after these revisions.
